# Trends in the Use of Non-Pharmaceutical Interventions in Schools During the COVID-19 Pandemic, February 2021 to December 2023: A Mixed Methods Study

**DOI:** 10.3390/ijerph22030394

**Published:** 2025-03-07

**Authors:** Nicole M. Robertson, Kailey Fischer, Iris Gutmanis, Veronica Zhu, Brenda L. Coleman

**Affiliations:** 1Sinai Health, 600 University Ave., Toronto, ON M5G 1X5, Canada; 2School of Public Health, University of Toronto, 155 College St., Toronto, ON M5T 3M7, Canada

**Keywords:** non-pharmaceutical interventions, COVID-19, education worker, hand hygiene, masking, mixed methods

## Abstract

The use of non-pharmaceutical interventions (NPIs) was imperative to avoid prolonged school closures during the COVID-19 pandemic. The purpose of this study was to understand the levels of adherence to and attitudes towards NPIs from February 2021 to December 2023 in schools in Ontario, Canada. Participants reported how frequently they, their coworkers, and their students used five NPIs: hand hygiene, covering coughs, staying home when ill, wearing a mask, and physically distancing. Open text comments provided participants with the option to provide additional details. Our mixed methods approach incorporated a series of descriptive statistics calculated at consecutive time points and thematic analysis. Participants reported higher adherence to NPIs than their coworkers and students, with less than perfect adherence that declined over time. Six themes emerged from the qualitative analysis on NPI use in schools: (1) the influence of time; (2) managing competing priorities; (3) a lack of enabling factors; (4) a lack of reinforcing factors; (5) the responsive use of NPIs; and (6) an emotional toll. To reduce the transmission of future communicable diseases and resultant staff and student sick days, ongoing commitment to hand hygiene, covering coughs, and staying home when ill is required.

## 1. Introduction

A novel strain of a coronavirus (2019-nCoV) was identified in Wuhan, China in December 2019 and later termed severe acute respiratory virus coronavirus 2 (SARS-CoV-2) [1]. Responses intended to disrupt the transmission of SARS-CoV-2 caused significant disruptions within the education system and contributed to learning losses, exacerbated educational inequalities, and changed the way we approach public education [2,3,4,5]. Non-pharmaceutical interventions (NPIs), measures to reduce the transmission of infectious agents that do not rely on vaccines or treatments, were used prior to, and as a complement to, vaccines against SARS-CoV-2 [6]. However, the use of NPIs in schools, and changes in their use over time, is ill-defined.

In Canada, the first case of the illness caused by SARS-CoV-2, coronavirus disease 2019 (COVID-19), was detected on 25 January 2020 [7]. The province of Ontario, Canada declared a state of emergency on 17 March 2020 [8]. Schools were closed until 30 June 2020, with remote learning instituted [9,10,11]. The following school year was also subject to temporary closures in response to heightened community transmission of SARS-CoV-2 [12]. In total, Ontario schools were closed for a minimum of 27 weeks, longer than any other Canadian province or territory [13]. Although school closures were only one of several mitigation strategies implemented to attempt to control the spread of SARS-CoV-2, it was acknowledged that they were detrimental to students’ learning and overall well-being [14,15].

No NPI is 100% effective at stopping the transmission of SARS-CoV-2 or other directly transmitted pathogens on its own. Also, some NPIs (e.g., lockdowns and school closures) are more socially disruptive than others. As such, the adoption of a suitable combination of NPIs, or layers of protection, is recommended to reduce the spread of infectious illnesses [16,17]. Although NPIs such as masking and physical distancing can be implemented temporarily in response to changes in rates of illness in the community or school, covering coughs/sneezes, hand hygiene, and staying home when ill are “routine” NPIs that should be practiced at all times, without regard to epidemics or pandemics [4,14].

To promote the safe reopening of schools in September 2020, the Government of Ontario proposed a layered approach emphasizing both routine (hand hygiene, covering coughs/sneezes, and staying home when ill), and temporary (mask wearing and physical distancing) NPIs [18]. Mask mandates were implemented in September 2020 for staff and students in grades 4–12 and extended to students in grades 1–3 in March 2021 [18,19]. The updated plan for the 2021–2022 school year maintained these requirements [19]. See Appendix A for a timeline.

Several studies that have largely focused on students have investigated the impacts of, or adherence to, NPIs in school settings during the COVID-19 pandemic [12,20,21,22,23,24,25,26,27,28,29,30]. Jarnig et al. [20] found that middle and high school students more frequently wore their masks correctly in the presence of teachers than in the presence of other students. Mickells et al. [30] reported that considerable effort by teachers was required to maintain adherence to masking in pre-kindergarten to second grade students. Lin et al. [28] presented reports from teachers on the challenging nature of monitoring students’ mask wearing, physical distancing, and hand hygiene. Amin-Chowdhury et al. [21] found that educational staff reported that both primary and secondary school students struggled with physical distancing and that hand hygiene was easier to implement in primary than in secondary schools. These findings suggest that education workers are already playing an important role in the promotion of NPIs among students. Given these challenges, a layered approach is needed to make up for failures of individual interventions.

Most studies of NPI use in schools have been cross-sectional in design. To date, few have investigated the temporal trends in their use, and none after 2022. The COVID-19 Cohort Study for Teachers and Education Workers (CCS-2) collected data from education workers between February 2021 and December 2023, a period that encompassed many scientific advances (e.g., vaccine development) and policy and attitude changes [31], including the rescinding of mask mandates in Ontario schools in March 2022 [32]. This study analyzes patterns of adherence to NPIs in the school setting throughout that period. In this study, we aimed to use both quantitative and qualitative methods to better understand the patterns of adherence to NPIs over time and to situate them within the context of the COVID-19 pandemic, as experienced in Ontario schools, through a pragmatic world view.

## 2. Materials and Methods

### 2.1. Design and Participants

This study was conducted as part of the CCS-2, a 35-month long prospective cohort study conducted with teachers and other education workers employed with elementary or secondary schools or school boards in Ontario [33]. This sub-study used a concurrent embedded mixed methods research design composed of a series of anonymized cross-sectional surveys, integrating both open- and closed-ended questions, completed annually over the duration of the study.

Recruitment began on 18 February 2021, following approval from the Research Ethics Board at Sinai Health. Rolling recruitment ended on 31 May 2023. Recruitment, consent, and data collection were conducted electronically due to COVID-19 restrictions on in-person research activities. Data collection ended upon the participants’ withdrawal or on 22 December 2023, whichever occurred first.

Participants were eligible for inclusion in these analyses if they were employed in any capacity by an Ontario public, Catholic, or independent school board; were 18–74 years old; and completed ≥50% of at least one baseline questionnaire. As participants could complete multiple baseline questionnaires during their tenure in the study, further inclusion criteria were applied to the individual questionnaires. Specifically, baseline questionnaires were not eligible for analysis if they were not submitted during the academic year in which they were assigned, if participants reported that they did not work in a school setting where students were present (i.e., they worked remotely, worked at a school board office, or were on leave/retired), or if a threshold of missing responses (as described below) was exceeded. Appendix A displays the numbers and reasons for exclusion of participants and reports.

### 2.2. Data Collection and Preparation

Following consent, participants were asked to complete a baseline questionnaire that captured all of the data used in these analyses. These included demographic characteristics, school/workplace characteristics, frequency of personal use of NPIs, and estimated use of NPIs by coworkers and students. An opportunity for general comments was also provided. Of note, an opportunity to provide open text comments was presented at the end of the questionnaire following the questions about NPIs (i.e., it was not specifically asking about NPI use). Participants were asked to complete an update of the questionnaire at the beginning of every new school year (September) to capture changes in information.

Quantitative analyses focused on the frequency of use of each of the five NPIs. Participants were asked the following questions: “While at work (since the beginning of the school year) how often do you… (1) physically distance from others, (2) wear a mask in others’ presence, (3) cover coughs, (4) wash hands thoroughly and regularly, and (5) stay home when you have symptoms, even if they are mild”. Participants were also asked “how often do your co-workers/students in your school…” use the NPIs listed above. Physical distancing was not defined in the questionnaire; the Ontario government defined physical distancing in the 2021–2022 school year as “as much distancing as possible” [18,19]. Responses were scored on a four-point Likert-like scale (never, occasionally, usually, or always) and were then dichotomized into always versus not always for statistical analyses since, for example, masks were required to be worn by staff and students (with reasonable exceptions) [19]. If more than two of five responses to NPI use were missing for any of the three groupings (self, coworkers, students), the questionnaire was excluded. Missing responses were not included in the denominators.

For quantitative analyses, baseline questionnaires were assigned to the academic half-year in which they were submitted (i.e., September to January or February to August). Although the questions asked respondents to estimate the use of NPIs “since the start of the school year”, responses were split into half-years to reduce the influence of recency bias.

### 2.3. Analyses

Mixed methods research involves philosophical assumptions and the integration of both quantitative and qualitative data analyses, thus improving the results of studies beyond what would be achieved by using either component alone [34,35]. A pragmatic worldview was used in these analyses, aimed at providing a practical solution in the context of the social, historical, political, and cultural background [34,36].

The quantitative data used to investigate patterns in the use of NPIs over time were supplemented with qualitative data on the attitudes and motivations behind the use of NPIs. Descriptive statistics were calculated for sociodemographic characteristics. The outcomes for each of the five NPIs included percentages and 95% confidence intervals calculated for each academic half-year and demonstrated graphically.

The open text responses were initially reviewed, and excerpts that were not applicable to NPI use were removed. The data were then analyzed using thematic analysis and reviewed to determine if behaviours and attitudes underwent a discernable shift during the data collection period. A hybrid approach of inductive and deductive coding, that integrated both data-driven and theory-driven codes, was used to develop an analytical codebook [37]. A preliminary codebook (deductive component) was developed by the first author (N.M.R.) based on the research objectives, the literature review, the patterns observed in our descriptive statistics, and a preliminary scan of the open text responses [37,38,39]. The initial codebook was adapted after consultation with K.F. and B.L.C.. An inductive approach was then applied to identify and capture previously unidentified themes [37,39,40], to enhance the themes identified in the deductive component, and to give voice to the education workers in our study [41]. After the first round of inductive coding, the research team reviewed and approved the proposed themes and subthemes; two rounds of inductive coding were completed by N.M.R. and B.L.C., who came to a consensus about any discordant coding. Data preparation and analyses were conducted using Stata/SE 18.5 [42], while the thematic analysis was conducted using Dedoose 9.2.22 [43].

## 3. Results

Of the 3876 education workers enrolled in the study, 3617 (93.3%) were eligible for inclusion in the quantitative data analysis; see Appendix A for specific exclusions. The mean age of the eligible education workers was 45.3 years, 3091 (85.5%) identified themselves as female, and 2923 (80.8%) as a teacher or instructor (see Table 1 for further details). Demographic details by each half-year period are available in Appendix A.

### 3.1. Quantitative Results: Reported Use over Time

The self-reported personal use of all five NPIs declined over the data collection period (February 2021 to December 2023), as did the estimated use of each by coworkers and students. Participants reported that they always adhered to each of the five NPIs more often than did their coworkers or the students attending their schools.

The largest decline in use of NPIs was for mask usage (Figure 1). In the first half year (spring 2021), when mask mandates were in effect, 79% of participants reported they always wore their masks while at work, as did 57% and 33% of their coworkers and students, respectively. These reported percentages dropped to 20%, 0.5%, and 0.1% for themselves, their coworkers, and their students, respectively, in the fall/winter 2022/2023 (the first data collection following the March 2022 lifting of the mask mandate). By the end of the data collection period (fall 2023), these estimates had dropped to 6%, 0.4%, and 0, respectively.

High percentages of participants reported that they always covered their coughs (≥88%) and washed their hands (≥68%) throughout the study period (Figure 2 and Figure 3); however, there was a ~10% drop from study start to end. The reported ability to always physically distance was low among participants (32% at the height) throughout the entire study period (Figure 4). The percentage of participants who reported always staying home when ill decreased by 50 points, from 75% to 26%, during the study (Figure 5), with similar declines reported for coworkers and students. The reported use of NPIs by school setting (elementary and secondary school) is available in Appendix A. The most notable differences by school setting were for students’ level of adherence to guidelines.

### 3.2. Qualitative Results: Themes

For open text responses, 945 comments from 718 participants were related to the use of NPIs in a school setting and were eligible for thematic analysis. The thematic analysis resulted in 6 themes and 14 subthemes, as illustrated in Figure 6. While some themes were omnipresent or recurrent over the duration of the data collection, other themes were more apparent during certain periods. Barriers to high levels of NPI use were common among comments.

#### 3.2.1. Theme 1: The Influence of Time

The influence of time was split into two subthemes: pandemic fatigue and a post-mandate behavioural shift. Pandemic fatigue referred to the changing landscape of the adoption of NPIs as the pandemic continued. Consistent with the close-ended question responses, participants described observing a “fatigue towards” or gradual waning in the use of NPIs over time among coworkers and students.


*“Over the past 3 months, fewer & fewer students are adhering to COVID protocols within my school; administration has become lax/no longer interested in enforcement.”*
(Teacher, secondary school, March 2021)


*“I can say that students have become less and less likely to use hand sanitizer regularly as the pandemic continues. Additionally, they are becoming more and more relaxed about masks and the new thing is to wear them under their noses.”*
(Teacher, mixed elementary and secondary school, April 2021)


*“The majority of staff and students comply but I have noticed a difference in the strict adherence to physically distancing and masking. They are more relaxed.”*
(Administrator, elementary school, December 2021)

The subtheme of a behavioural shift emerged after mask mandates were lifted in schools in March 2022; this was mirrored by the quantitative data, above. Participants noted a dramatic change in the use of, and attitudes towards the use of, NPIs, with a worsening since the start of the pandemic.


*“When masking requirements dropped many teachers and students at my schools stopped trying to avoid spreading illness.”*
(Teacher, elementary school, July 2022)


*“There was a big change in masking when mandate was dropped. Out of twenty students in June, there was only one that continued masking until the end of the school year. About half the staff (maybe as much as 75%) stopped masking when the mandate was removed.”*
(Teacher, elementary school, July 2022)


*“Due to the lifting of isolation requirements when ill, many in my school (including adults & students) still come to school when they have mild to moderate symptoms such as hacking coughs, runny noses, sore throats, fatigue..etc.”*
(Teacher, elementary school, October 2022)


*“While last year [September 2022–June 2023], the culture at school seemed to be that people stayed home if they were sick, even if symptoms were mild, this year the culture seems to have shifted, and more students and teachers are coming to school sick. Sometimes they mask; sometimes they don’t. The school has returned to pre-Covid rules regarding illness, so employees are required to bring a doctor’s note if they are absent for three days or more.”*
(Professional support role, elementary and secondary school, October 2023)

Other educators commented that their coworkers and/or students appeared to have dismissed fundamental transmission-limiting behaviours that were better than usual earlier in the pandemic.


*“Once the mask mandate was lifted, I really feel like people stopped covering their mouths when they coughed and people came to work sick again.”*
(Teacher, secondary school, August 2022)


*“It seems everything we learned during the pandemic about how to keep each other healthy has been lost.”*
(Teacher, secondary school, May 2023)

#### 3.2.2. Theme 2: Managing Competing Priorities in the School Setting

This theme encompassed priorities that conflicted with supporting the academic development and general well-being of students due to the need to adopt NPIs. The subtheme of job duties that hindered/conflicted with the adoption of NPIs refers to the day-to-day tasks that conflicted with the ability to consistently employ NPIs. It was more frequently mentioned regarding an inability to physically distance from young students and students with exceptionalities.

*“I teach students with special needs, so physically distancing myself is impossible. In order to provide the necessary support, I need to touch their hands, their learning materials, the food containers* etc.”(Teacher, secondary school, April 2021)


*“Close contact is required with unmasked children to assist with opening containers in lunches during nutrition break. Close contact is also required when assistance is required, for example dressing for outside, helping with shoe, packing or unpacking backpacks, and assisting with toileting accidents.”*
(Early childhood educator, elementary school, April 2021)

Other job duty-related exceptions, such as providing guidance and working with students with hearing loss, were noted to interfere with their and/or their students’ ability to always mask.


*“Physical distancing is tough with my job-guidance deals with private, often confidential information and sometimes upset and/or emotional students. I physical [sic] distance when I can. Sometimes panic attacks etc. (very prevalent now) require student to remove mask to breath and calm down.”*
(Teacher, secondary school, October 2021)


*“I want to wear a mask at work, but I cannot as I have students who read lips. They are hard of hearing.”*
(Teacher, secondary school, September 2023)

The subtheme of NPI use interfering with teaching/educational development referred specifically to the ability to proficiently deliver the curriculum to students and for students to be able to receive said instruction. Although comments within this subtheme appeared consistently across the data collection period, the use of different NPIs was mentioned more often at different times. Education workers were concerned that what they were expected to do to reduce the risk of COVID-19 in schools (masking, distancing) conflicted with what they understood to be beneficial to the learning and development of children.


*“Covid safety requirements in classrooms are not in line with “best practices” pedagogically-speaking; meaning, what we must do for our health/safety is in direct contrast to what we know is best for kids’ learning in K and Gr. 1. It’s challenging and stressful.”*
(Teacher, elementary school, August 2021)

Educators were also concerned about juggling priorities to monitor the use of NPIs in their setting.


*“There were many protocols we had to ensure in our schools (handwashing, hand sanitizing, monitoring physical distance, teaching and monitoring protocols) at the same time as being expected to teach all of the curriculum.”*
(Teacher, elementary school July 2021)


*“Getting students to wear their masks consistently and properly is an exhausting struggle”*
(Teacher, secondary school, December 2021)

They also noted that as the pandemic progressed, more children were coming to school when they were unwell.


*“Students are often coming to school sick; I have had several students who seem very ill in class but have come back to school because “they’ve missed too much already.”*
(Teacher, secondary school, October 2022)

#### 3.2.3. Theme 3: Lack of Enabling Factors

The term enabling factors was used to describe the conditions necessary for the adoption of NPIs, such as infrastructure, supplies, and human resources. The absence of a necessary condition would make it exceedingly difficult or impossible to adopt the corresponding NPI. The subthemes of infrastructure and a lack of supplies were consistently observed across the study period, while a lack of human resources was more common after the lifting of mask mandates. One of the more frequently reported barriers to successfully adopting NPIs was the lack of space to practice physical distancing.


*“We do not have enough space to physically distance in a school. This is impossible with class sizes. Let’s stop pretending this is a reality.”*
(Teacher, elementary school, March 2021)


*“I teach grade 2 students and it is very hard for them to remember to physically distance. It is also impossible for everyone in my class including myself to maintain physical distance as it is an overcrowded classroom of 21 students. There is only enough room for the desks in the class to be placed less than 1 m apart.”*
(Teacher, elementary school, March 2021)


*“Students would love to physically distance themselves from me, but there is no room in the classroom for them to do so. Class sizes are larger than they’ve ever been. Even before the pandemic.”*
(Teacher, elementary school, October 2021)

A lack of hand hygiene facilities was also noted by several participants.


*“I am a kindergarten teacher who was in person until the lockdowns. I was in a room with no sink or bathroom. I was left to only use hand sanitizer for washing hands.”*
(Teacher, elementary school, May 2021)


*“It is impossible to follow all rules in current school settings. Not enough sinks, washrooms to wash hands. One sink per 25 students if we are lucky. By the time hands are washed- a lesson is over.”*
(Teacher, elementary school, December 2022)


*“I’m in a portable. Time is tight and the washroom is far. Handwashing feels like a luxury.”*
(Teacher, elementary school, October 2023)

Some described a lack of necessary supplies such as warm water and soap/hand sanitizer, and/or proper fitting masks.


*“We have everyone sanitize as best we can- but often we are out of sanitizer and are looking for more.”*
(Educational assistant, elementary school, February 2021)


*“Students are required to wear masks but often the masks are ill-fitting and slip off their noses.”*
(Teacher, secondary school, April 2021)


*“The water available for the kids to wash their hands is ice cold. Truly ice cold. It is painful (even as an adult) to wash hands for a full 20 s.”*
(Teacher, elementary school, January 2022)


*“…there is no way to dry our hands—school cannot afford paper towel.”*
(Teacher, elementary school, October 2023)

After the lifting of mask mandates, the subtheme of a lack of human resources was identified as a major barrier to education workers staying home when ill. Many participants noted that there was heightened pressure from their administrators to be present despite being unwell, as there were not enough staff to cover absences.


*“The rules changed a lot over the academic year. No one stayed home unless they tested positive for covid. Education workers couldn’t stay home as there were often no supply/occasional workers to cover us, so we only stayed home if we had covid or if we were required to isolate as someone in our home had covid.“*
(Teacher, elementary school, July 2022)


*“We have almost no supply teachers so staying home makes it challenging for the rest of the school team. Many staff come in sick because they don’t want to strain a system that is already running on duct tape and hope.”*
(Teacher, elementary school, October 2022)


*“We have a severe supply teacher shortage so many, including myself, are coming to school sick because the impact on other staff and the students is so high when we can’t get a supply teacher! I taught 1.5 days last week when I should have been home sick for this reason!”*
(Teacher, elementary school, October 2022)

#### 3.2.4. Theme 4: Lack of Reinforcing Factors

The term reinforcing factors was used to describe conditions that strengthened or supported the ability to adopt NPIs in the school setting. These were further categorized as a lack of clear policies to support their adoption or policies that oppose adoption; a lack of school administrative support; and a lack of parental support. While all three subthemes were present for the full duration of the study, they were commented on more frequently as the pandemic continued.

Participants noted an absence of strict policies to support NPI adoption, policies that were difficult to interpret and thus difficult to adhere to, or those that conflicted with best practices for reducing COVID-19 transmission in schools. Staying home when ill was the NPI most often commented upon in this subtheme.


*“I have been directed to return to work on Monday if my result is negative even though Hospital Test Centre says I should isolate.”*
(Teacher, elementary school, March 2021)


*“There is much confusion from the families we teach about when to stay home and for how long. Very little clarity has been given by the board or our union.”*
(Teacher, elementary school, March 2021)


*“Very mixed messages between school boards and what Health Unit recommendations are in situations that cause stress.”*
(Educational assistant, elementary school, May 2021)


*“The [school board] staff ‘attendance program’ has also discouraged sick staff from staying home when ill, in fear, which also may impact medically fragile people.”*
(Teacher, elementary school, November 2023)

Participants also noted confusion regarding the use of masks after mandates were lifted. They also reported that guidelines for masking were frequently ignored by both staff and students.


*“We are doing the best, but without strict masking guidelines, students are happy to be mask free.”*
(Principal/vice principal, elementary school, October 2022)


*“Even staff who have been off with COVID do not mask upon their return as it is “guidance only.”*
(Teacher, elementary school, December 2022)


*“Staff and students do not generally stay home unless vomiting or feverish, and are returning from being sick and NOT wearing a mask.”*
(Teacher, elementary school, October 2023)

Several participants conveyed a perceived lack of support from their school’s administration when attempting to adopt NPIs or to monitor/enforce adherence to NPIs. Some participants noted a disinterest by their administration or senior leadership in sending children home when they were ill. Others noted a lack of role modeling and/or disregard for public health advice by senior leadership.


*“1. Many students are still being sent with symptoms. My principal will keep kids at school if she can and they are never placed in the isolation room. 2.When I said my coworkers and I usually wear masks, the only time we don’t is when eating in the staff room. We’re spread out there. There is however 2 non-teaching staff who don’t wear masks after school but our principal doesn’t think Covid is a big deal so it’s not dealt with”.*
(Teacher, elementary school, March 2021)


*“One of my students was sick with a cough and cold and on medication. He was crying and not able to cope with school. My principal would not send the student home.”*
(Teacher, elementary school, October 2021)


*“Our school board has decided that anything related to keeping staff or students safe is a “political issue” and I am quite literally the ONLY person wearing a respirator in my schools... Principals and system administrators refuse to wear a mask, even when clearly symptomatic.”*
(Teacher, elementary school, October 2023)

A lack of parental support was one of the more prevalent subthemes throughout the study. This was predominantly related to the ability to keep ill children home from school in the elementary school setting. Many participants noted that parents sent ill children to school despite policies in place to prevent such, while some described methods used to mitigate the appearance of symptoms that would result in being sent home.


*“Poverty and affordable child care affects families differently. Many of our families can’t afford to keep their kids at home or stay home from work if they are feeling unwell.”*
(Teacher, elementary school, July 2021)


*“MANY parents send the kids to school deliberately ill and noticeably sick (some with child care needs due to work and some at home parents) in December 2021 at least 1/2 of my class reported being given medication (the pink one- the bubblegum flavour one etc in the AM before school).”*
(Teacher, elementary school, February 2022)

Participants also reported an increase in the frequency of ill children being sent to school after the lifting of mask mandates.


*“Now that COVID protocols have been lifted, parents are sending their sick kids to school every day. Parents are even refusing to pick up their sick children.”*
(Teacher, elementary school, September 2022)


*“It appears that parents believe COVID is over and this year [2022,2023] so many kids are coming to school very sick.”*
(Teacher, elementary school, October 2022)

#### 3.2.5. Theme 5: Responsive Use of NPIs

The responsive use of NPIs referred to the addition, substitution, or adaptation of NPIs in response to other influences. The first subtheme, the addition/substitution of NPIs, was mentioned throughout the duration of the study. Participants said they used an additional layer of protection to mitigate instances where the use of another NPI was not possible or satisfactory.


*“If we need to be closer to them (to give instructions or explain paperwork), then I always wear a face shield on top of my mask.”*
(Teacher, secondary school, March 2021)


*“I wear my mask all the time and eat by myself. I keep windows open all the time.”*
(Teacher, elementary school, October 2022)


*“Last year 99% of my students masked all day. This year 4/27 mask. We have a HEPA filter running in the middle of the classroom. I religiously turn it up to max when they are gone and keep 2+ windows open throughout the entire day.”*
(Teacher, elementary school, February 2023)

Some education workers proposed additional measures that were not supplied by the schools that they felt would help overcome difficulties.


*“I use a microphone, but teachers and both administrators (principal and VP) frequently hold their masks off of their faces to project into the room. All should be issued microphones to discourage this behaviour.”*
(Teacher, elementary school, October 2021)

The subtheme of adapting NPIs became prominent after the lifting of mask mandates in March 2022. The most common adaptation was the use of a mask when most others were not using them.


*“I have noticed an uptick in students who are off sick returning and wearing a mask for a few days after their return... the same goes for staff members... I am not currently masking, though I plan to begin again during cold and flu season, especially before I receive my flu shot.”*
(Teacher, secondary school, September 2023)


*“I have worn a mask and distanced at school when positive for COVID. Once I tested negative on RAT test and symptoms subsided, I stopped wearing a mask at work.”*
(Teacher, elementary school, September 2023)

#### 3.2.6. Theme 6: Emotional Toll

The sixth theme that emerged was the emotional toll experienced by education workers and their students during the COVID-19 pandemic as it related to the adoption (or lack thereof) of NPIs in schools. Two subthemes emerged: emotional distress related to the adoption of NPIs, and distress related to the lack of adoption of NPIs. Both subthemes emerged as adherence to NPIs was waning. Participants noted that the ongoing use of NPIs was perceived negatively by coworkers and students and often resulted in the users feeling isolated.


*“I used to wear my mask regularly last year but this year I feel some social pressure to not wear the mask as much.”*
(Teacher, secondary school, October 2023)


*“I am the only masking member of staff, and it is deeply unpopular with my colleagues, who talk about my “paranoia” and speculate, publicly, about my mental health.”*
(Teacher, secondary school, October 2023)

Given the widespread lack of mask usage after the lifting of the mandate, participants noted that their use of masks signified illness and infectiousness to peers, causing fear.


*“I stayed home 3 days (+2 days from the weekend), because the internet said it was less contagious after 5 days, and wore a mask the first day back, but that alarmed students because they thought I had COVID, so I didn’t wear a mask the Friday.”*
(Teacher, secondary school, October 2023)

Alternatively, participants conveyed distress associated with the lack of NPI use; this was more frequent as NPI use waned.


*“We have no choice but to risk health for employment. Many in schools refuse to follow public health measures.”*
(Teacher, secondary school, March 2021)


*“Very few other teachers or students wearing masks at school. Lots of people coming into school when sick, it feels very dangerous and stressful.”*
(Teacher, elementary school, October 2022)


*“Admin does NOT wear masks, they tell us Covid is over!?? So many students & staff are continually sick & often don’t stay home. I am seeing already that Long Covid will be the reality for many staff & students 2, 3 + reinfections of Covid seems to be the norm. I am living a nightmare everyday—no-one seems worried or concerned. I am terrified for the future.”*
(Teacher, elementary school, February 2023)

Some conveyed dismay that little was being done to protect staff and students and their right to a safe learning environment.


*“It seems so bizarre to me that with the 6th wave here and being recommended to continue wearing masks (indoors with people) that schools and people inside no longer are mandated to do so? I do not feel safe in this environment!”*
(Educational assistant, secondary school, April 2022)


*“The complete disregard for the health and safety of the medically vulnerable (including myself) during this time of the pandemic has been disheartening and a complete failure of the provincial government and public health.”*
(Teacher, elementary school, January 2023)

## 4. Discussion

This study of elementary and secondary school education workers in Ontario, Canada described the use of, and experiences associated with, the adoption of NPIs in schools from February 2021 to December 2023. Quantitative analyses demonstrated that the use of NPIs (i.e., masking, physical distancing, hand hygiene, covering coughs/sneezes, and staying home when ill) was less than ideal and trends in their use declined over time among education workers, their coworkers, and their students. Barriers to the use of NPIs included time/priority setting; the lack of infrastructure, human resources, supplies, and supportive policies; and a lack of support from school administration and/or parents.

Three areas of opportunity for improvement were raised from our analyses: (1) reducing barriers that impede the ability to adopt NPIs among education workers and students; (2) the necessity of using and supporting the use of hand hygiene, covering coughs/sneezes, and staying home when ill on a permanent basis; and (3) the need to institute the use of NPIs, such as masks, on a short-term basis during outbreaks/periods of increased illness.

The establishment and continued reinforcement of policies in which a workplace safety climate is promoted would reduce the barriers to the use of NPIs by education workers and the promotion of NPI use to their students/coworkers. The concept of a safety climate refers to the perceptions that employees share about the many aspects of safety within their working environment [44,45]. Education workers in this study frequently noted the need for policies and/or the enforcement of existing policies to increase the use of NPIs. Likewise, in a 2020–2021 study, Serrano et al. [46] reported that education workers in Ontario felt that their work environment was inadequately safe and that students and staff were unable to fully follow COVID-19 protocols. A study conducted in the same era (December 2020) found that two-thirds of almost 5000 educators in Ontario felt that they had less than half of the needed infection control protocols in place and noted deficits in physical distancing measures, screening students for illness, and masking requirements for students [47]. In a separate survey among the education workers in this cohort, conducted in December 2022, similar sentiments were reported, with over 50% of comments suggesting that masks should be mandatory for all staff and students and 25% stating that principals and superintendents enforce the stay-at-home rules [33]. The establishment of workplace safety climates in education systems may help address many concerns, including concerns about infectious disease transmission.

Routine personal protective measures such as hand hygiene, covering coughs/sneezes, and staying home when ill are an integral part of preventing illnesses caused by infectious agents. They are proven strategies to prevent illness that will continue to be important beyond the end of the COVID-19 pandemic [48]. In the school setting, the optimization of the health and safety of education workers and students requires that these routine measures are in place and are supplemented with other measures such as adequate ventilation, environmental cleaning, and vaccination [4,49]. Hand hygiene and covering coughs and sneezes are core prevention strategies to limit the spread of infections that are modifiable. School-based and early education programmes that promote hygiene can reduce the incidence of COVID-19 and other infections that circulate within schools and thereby reduce illness-related absenteeism [49,50,51]. The early formation of hygienic habits has been associated with higher adherence to pandemic preventive practices among adolescents, suggesting that early intervention also promotes sustainability in hygienic habits [52,53]. School staff can teach and reinforce good hygiene to students, but they must be equipped with the necessary tools to do so. In this study, barriers to the use of hand hygiene included a lack of infrastructure and/or supplies. A programme that supports education workers to model and teach hygiene according to the children’s developmental stages would be beneficial [54], as would policies to ensure that schools have the facilities and supplies to support them.

Steps need to be taken to reduce presenteeism (i.e., attending school/work while ill). Presenteeism increases morbidity and diminishes the productivity and quality of work of education workers [55,56], and ultimately negatively impacts the educational attainment and mental and physical health of students [57]. There is a large body of evidence to support school and workplace exclusion policies [58,59,60]. However, the prolonged nature of the COVID-19 pandemic introduced new considerations for absenteeism in the school setting (i.e., learning losses). Study participants described the lack of human resources as a strong driving force behind feeling supported with the decision to stay home when ill. This issue requires a two-fold solution: reducing worker productivity losses associated with absenteeism among currently employed education workers by identifying and reducing risk factors for illness [55] and assuring adequate human resources to replace workers who need to stay home when ill [61].

It is understood that allowing ill students to attend school perpetuates the spread of infections [62,63]. Further, studies among university students found that students attending classes while ill frequently reported difficulty concentrating in class, being tired and distracted, and studying slower [64] while having a negative impact on general health, well-being, and the overall ability to study [65]. Participants in this study noted several barriers that prevented students from staying home when ill, many of which extended beyond the classroom/school system. The most frequently cited barrier was a lack of childcare/the inability of a parent to stay home with an ill child. A systematic review highlighted factors in parents’ decisions to send children to school while ill; these included not recognizing symptoms of illness as well as factors external to illness (e.g., the school’s sick policy), the child’s attendance history, and the ability to arrange childcare and/or to stay home themselves [57]. That review also noted that parents used medications to mask symptoms of illness in their children, a behaviour also mentioned by our study’s participants. It is important to have robust measures in place to reduce illnesses (e.g., use of NPIs) while supporting absences due to illness [57], including clear and consistent communication among the involved parties, reinforced by the fair and equitable enforcement of existing policies [51].

Wearing a mask and physical distancing can be challenging to implement and may impact social connectedness in the school setting [4]. Despite this, they remain important NPIs that can be used as part of a layered approach to infection prevention. The re-introduction of these measures may be beneficial in periods of increased transmission of respiratory infections within the school and the wider community. With the lifting of mask mandates, wearing a mask in school remained a personal choice that was influenced by complex social norms, peer influence, and local policies [66,67]. In December 2022, calls to re-introduce widespread masking in schools occurred as surges in seasonal respiratory illnesses, such as influenza and respiratory syncytial virus, occurred along with ongoing SARS-CoV-2 transmission. This caused the overcrowding of children’s hospitals across Canada [68]. And yet, participants in this study reported that the majority of staff and students chose not to mask when given the choice, even when masks were strongly recommended (e.g., when symptomatic or recovering from COVID-19). This may be indicative of the ongoing politicization of mask wearing in North America [69,70]. Study participants who chose to continue to wear their masks or wore them when symptomatic noted feelings of isolation and judgement from their peers. Further research on how best to support those who choose to wear masks in schools (and elsewhere) and how to successfully re-introduce masking during periods of heightened transmission of respiratory viruses is necessary. That research needs to be attentive to social and behavioural norms.

Physical distancing was the most poorly performed NPI in this study. Participants reported that attempts to distance were quickly abandoned and that they felt frustrated over the expectation to physically distance while not being able to do so. In addition to the lack of space required to physically distance in the classroom, participants also noted instances when distancing was not well understood by young students and of job duties interfering with their ability to distance. Others noted that group work and social interactions important for learning and development were not possible if practicing physical distancing. Given this, other layers/strategies to reduce transmission should be prioritized over physical distancing in school settings.

Participants alluded to both a pandemic fatigue, described by the World Health Organization [71] as the “demotivation to follow recommended protective behaviours, emerging gradually over time and affected by a number of emotions, experiences and perceptions” (p. 4), and a general shift in adherence after the lifting of mask mandates in Ontario. The decline in the percentage reporting they, their coworkers, and their students always used NPIs supports this. Several longitudinal studies investigated the concept of pandemic fatigue with reported reductions in the use of NPIs from early to late 2020 [72,73,74,75,76,77,78]. The reduction in the use of NPIs continued into 2021, with a report of declines in mask wearing in the USA in states that both did and did not lift state-issued mask mandates in early 2021 [79]. Similarly, the rates of hand hygiene, mask wearing, and physical distancing declined between April and December 2021 in Belgium [80], and there was a significant decline in rates of mask wearing while at work among UK participants between May and November 2022 despite the ongoing transmission of SARS-CoV-2 [81]. In contrast, Brankston et al. [72] found increasing precautionary behaviours in the context of increasing disease incidence between June and November 2020 in Canada. A study conducted between early 2020 and July 2022 in the Netherlands similarly reported that adherence to guidelines about hand hygiene and physical distancing fluctuated in response to rates of hospitalization due to COVID-19 [82]. Of interest, these authors reported a decrease in the rates of hand hygiene over the study period but no change in distancing behaviours. Our study observed an ongoing fatigue in all measures, both routine and temporary. COVID-19 recovery actions should give attention to reinvigorating the use of routine NPIs to reduce the transmission of infectious diseases at all times and consider the use of additional temporary NPIs during outbreaks/periods of increased disease incidence.

A gradual easing of restrictions caused by COVID-19 commenced after the peak of the Omicron BA.1 wave, and culminated with dropping mask mandates in Ontario schools (and most other public settings) in March 2022 [32]. Participants attributed the abrupt decline in NPI use to the lifting of mask mandates, with several describing attitudes or behaviours that may have signified the widespread belief that the COVID-19 pandemic had ended. Authors of other studies have suggested that policies that people perceive as restrictive and/or polarizing, such as mask and vaccine mandates, may result in a decline in trust in governments and scientific institutions and a reduction in the uptake of future public health measures [83]. While the arguments for or against the use of mandates and other restrictive measures are beyond the scope of this paper, it would be amiss to ignore the consequences that may have resulted from them. As such, a multidisciplinary approach that includes scientists, child educators, public health officials, and government officials is required to reestablish the importance and utility of NPIs for the prevention of infectious agents beyond COVID-19.

## 5. Study Strengths and Limitations

The data for this study were collected over almost three years, enabling the review of trends over time, including the lifting of mask mandates in the Ontario school system. This study includes many occupations within the school setting with differing levels of exposure to staff and students, providing a more complete picture than those that only capture teaching roles. To our knowledge, our study is the first longitudinal study to describe pandemic fatigue of both routine and temporary NPIs among education workers and students.

This study relied on voluntary participation, making it unlikely that participants were representative of all education workers. However, participants were recruited from across the province among both public and private school systems, making findings more generalizable to the whole of Ontario and to jurisdictions outside of the province and country. Also, the dynamic nature of participation in the study increased the potential for differences among participants over time. It also, however, allowed for the inclusion of participants throughout the data collection period without a sudden drop-off due to participation fatigue. Given the ad hoc decision to use open text comments for thematic analysis, we acknowledge that the responses may not be representative of all participants in the study or of all education workers in Ontario. It is likely that those providing comments were different in some way than those who did not. Despite this, giving voice to study participants is important since they have firsthand knowledge and experiences that cannot be understated or replicated. We also acknowledge the potential for bias in the coding and interpretation of the qualitative data given the authors’ backgrounds in public health and infectious diseases. Finally, despite the use of anonymized surveys, it is possible that social desirability bias resulted in inflated estimates of the personal use of NPIs.

## 6. Conclusions

The results of our study indicated that the use of NPIs during the COVID-19 pandemic was less than ideal in schools. Reasons gleaned from the data, and possible areas for improvement, included the need to manage competing priorities, a lack of enabling factors (infrastructure, supplies, and human resources), and a lack of reinforcing factors (clear policies and administrative and parental support). Pandemic fatigue and a behavioural shift following the revocation of mask mandates were likely the driving forces behind waning adherence to NPIs during this study.

## Figures and Tables

**Figure 1 ijerph-22-00394-f001:**
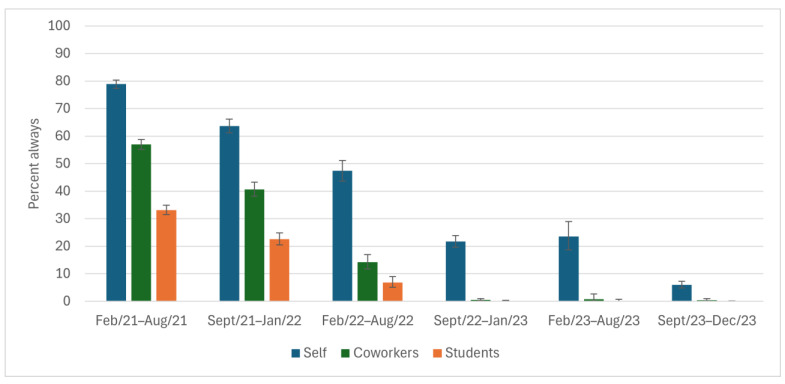
Wears a mask in others’ presence while at work, as reported by Ontario education workers for themselves, their coworkers, and their students; COVID-19 Cohort Study for Teachers and Education Workers (February 2021 to December 2023). The vertical bars indicate 95% confidence intervals.

**Figure 2 ijerph-22-00394-f002:**
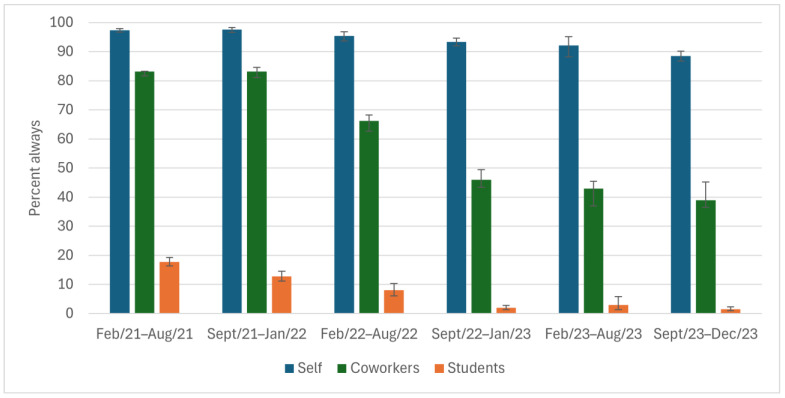
Covers coughs while at work, as reported by Ontario education workers for themselves, their coworkers, and their students; COVID-19 Cohort Study for Teachers and Education Workers (February 2021 to December 2023). The bars indicate 95% confidence intervals.

**Figure 3 ijerph-22-00394-f003:**
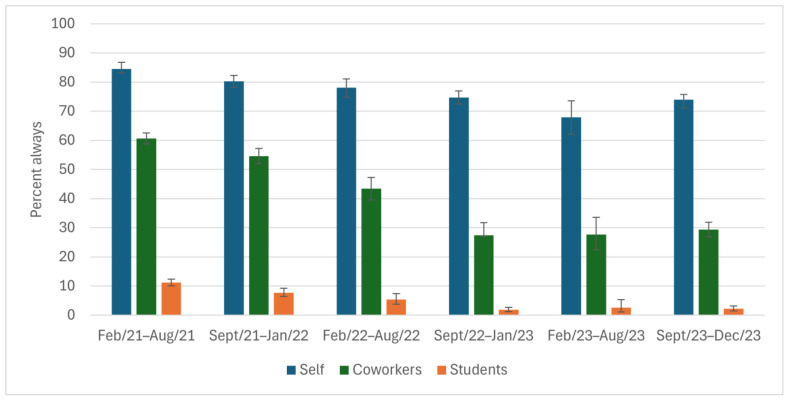
Washes hands thoroughly and regularly while at work, as reported by Ontario education workers for themselves, their coworkers, and their students; COVID-19 Cohort Study for Teachers and Education Workers (February 2021 to December 2023). The bars indicate 95% confidence intervals.

**Figure 4 ijerph-22-00394-f004:**
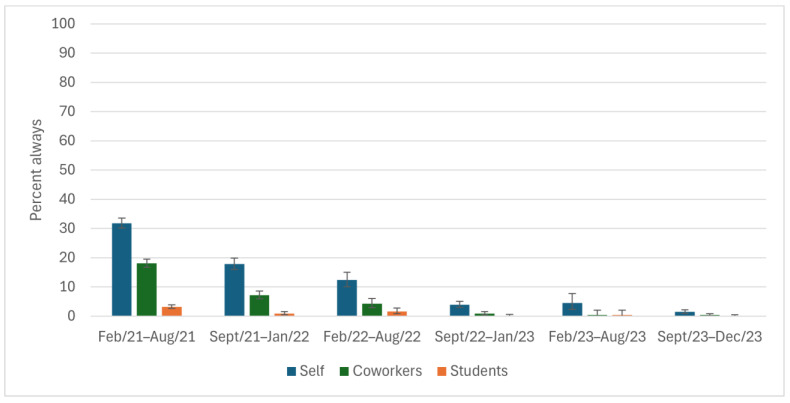
Physically distances from others while at work, as reported by Ontario education workers for themselves, their coworkers, and their students; COVID-19 Cohort Study for Teachers and Education Workers (February 2021 to December 2023). The bars indicate 95% confidence intervals. A specific distance was not included in the question.

**Figure 5 ijerph-22-00394-f005:**
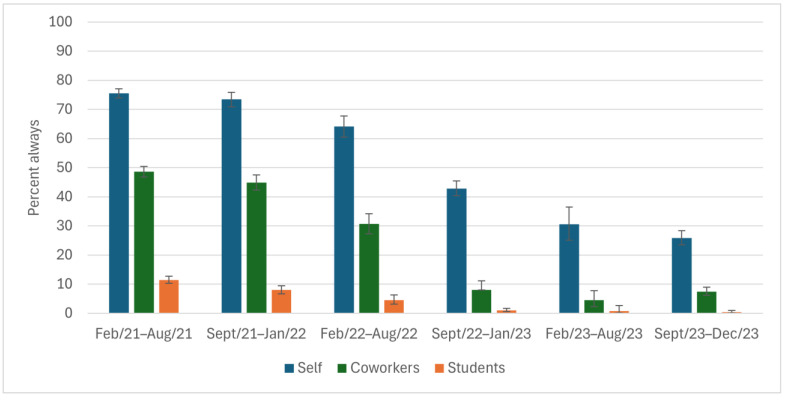
Stays home from work when they have symptoms, even if they are mild, as reported by Ontario education workers for themselves, their coworkers, and their students; COVID-19 Cohort Study for Teachers and Education Workers (February 2021 to December 2023). The bars indicate 95% confidence intervals.

**Figure 6 ijerph-22-00394-f006:**
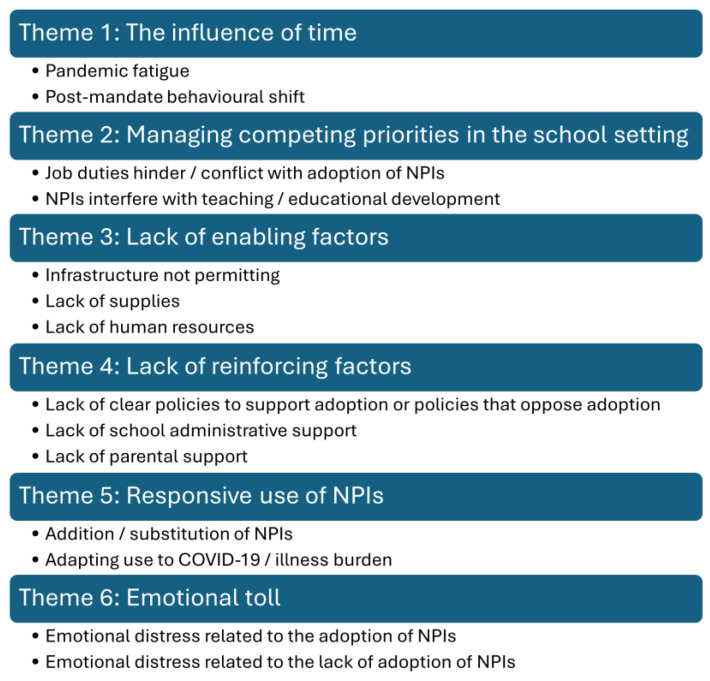
Thematic map with six themes and their subthemes; COVID-19 Cohort Study for Teachers and Education Workers; Ontario, Canada; February 2021 to December 2023.

**Table 1 ijerph-22-00394-t001:** Characteristics of Ontario education workers at the time of their first baseline report; COVID-19 Cohort Study for Teachers and Education Workers (February 2021 to December 2023). Number (percent) unless otherwise noted.

Characteristic	Participants N = 3617
Age in years, mean (95% CI)	45.3 (45.0, 46.0)
Gender	
Female	3091 (85.5)
Male	517 (14.3)
Other	9 (0.2)
Education, highest achieved	
Diploma, college, or less	341 (9.4)
Bachelor’s degree/teaching certification	2441 (67.5)
Graduate school (master’s or PhD)	835 (23.1)
Occupation	
Teacher	2923 (80.8)
Educational assistant	224 (6.2)
Early childhood educator	80 (2.2)
Principal/vice principal	133 (3.7)
Administration ^1^	87 (2.4)
Professional student services roles ^2^	130 (3.6)
Support staff ^3^	40 (1.1)
Chronic illness ^4^	
Yes	921 (25.5)
No	2696 (74.5)
Postal district	
Eastern Ontario	643 (17.8)
Central Ontario	1251 (34.6)
Metropolitan Toronto	721 (19.9)
Southwestern Ontario	831 (23.0)
Northern Ontario	171 (4.7)
School setting	
Elementary	2227 (61.6)
Secondary	1103 (30.5)
Both/mixed setting	287 (7.9)

CI: confidence interval. ^1^ Office/clerical staff, superintendents. ^2^ Psychologist, social worker, therapist, librarian, nurse. ^3^ Technicians, bus drivers, custodians, building maintenance, cafeteria staff, lunchroom assistants. ^4^ Asthma, chronic obstructive pulmonary disease or other chronic lung condition, diabetes, heart disease, cancer treated in the past five years, liver or kidney disease, HIV/AIDS or other immune suppressing disease/condition, chronic neurological disorder, or other long-term chronic conditions.

## Data Availability

The datasets generated and/or analyzed during the current study are not publicly available due to information that could compromise the privacy of research participants but are available from the corresponding author on reasonable request.

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
