# Peer review of "Trends in the Use of Non-Pharmaceutical Interventions in Schools During the COVID-19 Pandemic, February 2021 to December 2023: A Mixed Methods Study"

_ijerph, 2025, doi:10.3390/ijerph22030394_

Round 1

Reviewer 1 Report

Comments and Suggestions for Authors

This study focuses on the use of non - pharmaceutical interventions (NPIs) in Ontario schools during the COVID - 19 pandemic from February 2021 to December 2023. By using a mixed - methods approach, it comprehensively explores the adherence levels and attitudes towards NPIs among education workers, their colleagues, and students. The research topic is highly relevant to the public health and education sectors during the pandemic, aiming to provide valuable insights into school - based infection prevention and control measures.

The article is well - structured with a clear logical flow. The Introduction section effectively sets the stage by presenting the background of the COVID - 19 pandemic, the importance of NPIs in schools, and the research gap. The Materials and Methods section details the study design, participant recruitment, data collection, and analysis methods, ensuring the scientific rigor of the research. The Results section presents quantitative and qualitative findings separately, with the use of figures and themes to enhance the comprehensibility. The Discussion section deeply analyzes the results, discusses the implications, and proposes improvement directions. Finally, the study strengths and limitations are objectively presented, which is a comprehensive and systematic way of reporting research results.

The language used in the paper is generally fluent, with few grammar errors. The authors use appropriate scientific and technical terms accurately, which can clearly convey the research ideas and results. The writing style conforms to the academic norms, making it easy for readers to understand the content.

This manuscript presents a valuable mixed - methods study that explores the use of non - pharmaceutical interventions (NPIs) in Ontario schools during the COVID - 19 pandemic. The research fills an important gap in the literature by investigating the temporal trends of NPI use and the associated attitudes, which is crucial for understanding the effectiveness of these measures in the school setting. The study design combines quantitative and qualitative data, providing a comprehensive view of the topic. However, there are several areas that need to be addressed to enhance the quality and impact of the paper.

Areas for Improvement:

  1. Introduction

The introduction provides a thorough background on the COVID-19 pandemic and the importance of NPIs. However, it could benefit from a clearer statement of the study's specific objectives and hypotheses early on. This would help guide readers through the study more effectively.

  1. Methods

The recruitment process and data collection methods are well-described. However, the criteria for excluding participants could be explained more clearly to ensure transparency; The statistical methods used for analysis are appropriate, but the manuscript could benefit from a brief explanation of why the chosen statistical tests were most suitable for the data.

  1. Results

The figures and tables are clear and effectively illustrate the trends in NPI use over time. However, some of the figures(e.g., Figure 4)could benefit from additional context or explanation in the text to help readers understand the significance of the findings; The qualitative themes are well-developed, but the manuscript could provide more examples from the open-text comments to support each theme. This would give readers a better sense of the data driving the conclusions.

  1. Discussion

The discussion section effectively summarizes the key findings and their implications. However, it could benefit from a more detailed comparison with other studies, particularly those that have investigated similar topics in different regions or settings; The manuscript could also discuss the potential limitations of the study in more depth, such as the potential for social desirability bias in self-reported data.

  1. Grammar and Style

There are a few minor grammatical errors and awkward phrasings throughout the manuscript. For example, in the abstract, "The outcome for the quantitative analysis was the use of each of the five NPIs during the school year in which the baseline report was assigned" could be rephrased for clarity; Some sentences are overly complex and could be simplified to improve readability. For example, in the discussion section, "Steps need to be taken to reduce presenteeism(i.e., attending school/work while ill)because it increases morbidity and diminishes productivity and the quality of work of education workers" could be broken into two sentences for clarity.

  1. References

Ensure that all references are cited correctly and consistently formatted. There are a few instances where the citation style appears to vary slightly.

Comments on the Quality of English Language
  1. Overall Assessment

The English language in this manuscript is generally of a good standard, which is appropriate for an academic publication. The authors have a good command of scientific and technical language relevant to the field, and they are able to convey their ideas clearly in most cases. However, there are some areas where the language could be improved to enhance the readability and precision of the paper.

  1. Positive Aspects

Technical Language: The use of technical terms related to COVID - 19, non - pharmaceutical interventions, and educational research is accurate and appropriate. The authors successfully integrate these terms into the text without overcomplicating the narrative, which is a positive aspect for communicating with the target academic audience.

Sentence Structure: The sentence structures are diverse enough to convey complex ideas. For example, in the methods section, the authors use a series of well - formed sentences to describe the research design, data collection, and analysis procedures. This shows an ability to organize thoughts effectively in writing.

  1. Areas for Improvement

Clarity: Some sentences could be rephrased to be more straightforward. As mentioned earlier, certain complex sentence structures make the meaning less clear. Simplifying these sentences would improve the overall readability, especially for readers who may not be familiar with the specific jargon or complex concepts in the field.

Consistency: There is a lack of consistency in the use of abbreviations and terminology. This can be distracting for the reader and may cause confusion, especially when trying to follow the argument or results of the study.

Grammar and Punctuation: Although the number of grammar and punctuation errors is relatively small, they still exist. These errors, if left uncorrected, can detract from the overall professionalism of the paper. A more careful proofreading would help to eliminate these minor but noticeable issues.

Reviewer 2 Report

Comments and Suggestions for Authors
  1. Clarify the justification for dichotomizing Likert-scale responses (always vs. not always) in the Methods section. Would an alternative approach (e.g., ordinal regression) have provided more nuance?
  2. Expand the discussion on pandemic fatigue by comparing findings with similar studies from other countries. Were similar trends observed elsewhere?
  3. The qualitative thematic analysis is strong, but the coding process would benefit from additional details on inter-coder agreement and potential biases.
  4. The study mentions barriers to staying home when sick due to teacher shortages. Could policy recommendations address how schools might mitigate this issue?
  5. Consider including a supplementary table that shows how participant characteristics changed over time (e.g., were later respondents demographically different from early respondents?)
